# Effect of Mineral–Microbial Deodorizing Preparation on the Value of Poultry Manure as Soil Amendment

**DOI:** 10.3390/ijerph192416639

**Published:** 2022-12-11

**Authors:** Andrzej Cezary Żołnowski, Tadeusz Bakuła, Elżbieta Rolka, Andrzej Klasa

**Affiliations:** 1Department of Agricultural and Environmental Chemistry, Faculty of Agriculture and Forestry, University of Warmia and Mazury in Olsztyn, 10-718 Olsztyn, Poland; 2Department of Veterinary Prevention and Feed Hygiene, Faculty of Veterinary Medicine, University of Warmia and Mazury in Olsztyn, 10-718 Olsztyn, Poland

**Keywords:** poultry manures, mineral microbial deodorizing preparation, soil properties, plants yield

## Abstract

Poultry farming involves the production of poultry manures (PMs), which, if properly managed, are excellent organic soil amendments. Poultry farms generally do not have adequate arable land, and therefore, valuable fertilizer becomes a problematic waste. During the production and storage of PMs, odorous VOCs, NH_4_, H_2_S, and potent greenhouse gases such as CH_4_, CO_2_ are emitted. It influences the productivity of poultry and negatively affects the working conditions of working staff. In the present study, mineral–microbial deodorizing preparations (MMDP) based on perlite and bentonite as well as the following microorganism strains *Lactobacillus plantarum*, *Leuconostoc mesenteroides*, *Bacillus megaterium*, *B. subtilis*, and *Pseudomonas fluorescens* were added to the litter of turkey broilers (TB) and egg-laying hens (LH). PMs were compared with treatments without the addition of MMDP, and maize, sunflower, and rapeseed forage crops were tested. The influence on soil parameters such as pH, EC, HAC, SBC, CEC, BS, N_tot_, C_tot_, and plant yield and parameter of photosynthesis, i.e., SPAD index, was tested. Soil amending with manure resulted in an increase in pH and a decrease in HAC; in addition, an increase in EC, which was counteracted by the addition of MMDP, was noted. MMDP positively affected parameters such as SBC, CEC, and BS. It was shown that PMs, with the addition of MMDP, improved crops’ yield in the first year of the study, whereas this effect was not seen for the after-crop plants (lupine). The main ‘added value’ related to the usage of MMDP in poultry production is the improvement in the properties of PMs, which mainly had a positive effect on soil indicators.

## 1. Introduction

Poultry manure (PM) is a valuable natural fertilizer because it is a rich source of macro- and micronutrients for crops [1,2]. The use of manure reduces soil degradation and improves soil fertility [1,3] as well as the physical properties of the soil [1,2,4]. The long-term application of manure improves the water retention of soil [3,4,5,6] and increases soil temperature [4,7]. The use of PM increases soil pH and decreases hydrolytic acidity (HAC) [4], and increases the sum of base cations (SBC) and the cation exchange capacity (CEC) of the soil, thus, increasing the content of available nutrients [2,7]. The application of this soil amendment promotes the accumulation of soil carbon and accelerates the rate of nutrient turnover [3], providing available organic substances and nutrients, which increases the activity of soil enzymes [1,2].

The positive effect of using PM was also reported as better growth, and thus, a higher yield of crops [1,4,5,6,7,8,9,10,11]. As a result of the application of PM, an increase in the concentration of macroelements [2,4,5] and dry matter content in plant tissues [2] was observed [2] as well as changes in the leaf greenness index (SPAD), which, determined at critical stages of plant development, is considered a reliable predictor of green mass yield. The greenness index of leaves is determined by the availability of nutrients—mainly nitrogen, as well as thermal and humidity conditions [12], therefore, it should be assumed that the application of the PMs, of which determines the change in these conditions, will correlate with the SPAD index. Overdosing of PM may lead to over-saturation of the soil, which may be a factor limiting the nutrient uptake by the plant [8,13]. Some studies showed a positive effect of PM on the physical and chemical properties of soil as a result of its combined use with biochar [6,14,15].

The production of PM is an inevitable component of poultry farming, resulting in the emission of undesirable gas emissions from volatile odorous organic compounds and methane, carbon dioxide, ammonia, hydrogen sulfide and solid particles into the atmosphere [16,17,18]. This emission may be aggravated by the inadequate management of poultry manure [17]. To reduce the above-mentioned emissions from poultry manure, various types of deodorizing additives are applied to the litter, including mineral–microbial deodorizing preparation (MMDP) [16,19] or plant extracts [19]. Preparations such as MMDP are usually composed with the carriers of microorganisms—perlite or bentonite [19]. They, besides from fulfilling this function, can also increase the sorption capacity of these preparations to the components of manure. In agricultural practice, mineral additives such as perlite and bentonite have been used for a long time to improve soil sorption properties of soils [20]. Perlite is a naturally occurring silica volcanic rock composed mainly of the oxides of: silicon, aluminum, sodium, potassium, iron, magnesium, and calcium [21]. Bentonite is a sedimentary clay rock composed mainly of montmorillonite [22,23]. Its chemical composition includes mainly silicon with iron oxides and small amounts of Mg, K, and Ca [24]. Due to its high sorption capacity, bentonite is also used as a carrier for chemical fertilizers. By binding the ingredients, it is applied for the production of slow-release fertilizers [25], which means that the release of available nutrients is balanced with the needs of the crops [23]. Regarding bentonite, the literature indicates that it has a positive effect on the availability and retention of water in the soil. This mineral has a positive effect on the pH value of the soil, the capacity of the sorption complex, and it increases the content in the soil of total N, available Mg, P, and K, as well as total Ca, Mg, Zn, Mn, and organic C [22,23,26]. Furthermore, as a result of the application of bentonite to the soil, an increase in the content of N, P, and K in wheat and straw and an increase in the yield of millet was observed by increasing the weight of 1000 grains and the number of panicles [23,26]. The available data indicate that poultry manure enriched with MMDP should have a positive effect on soil properties and crop yields, but the minerals used in Deodoric^®^ may also, due to high sorption capacity, limit the availability of nutrients such as K and P for plants, as indicated by research conducted by Sager et al. [27]. The cited literature indicates that the development of effective MMDPs that reduce the emission of odorous volatile organic compounds and gases such as methane, carbon dioxide, ammonia, and hydrogen sulfide is the first problem. The second problem is the answer to the question regarding the influence of MMDP on the properties of manure and its impact on soil properties, the availability of plant nutrients and plant yield.

The influence of mineral–microbiological preparations on the quality of manure and its subsequent impact on soil and plants has not been fully explained in the literature to date; therefore, the research hypothesis adopted here assumes that MMDP can change the properties of manure and can significantly modify the properties of soils and the crops’ yield.

The aim of the investigation was to evaluate the influence of poultry manure with the addition of MMDP on the selected soil properties. This impact was analyzed in relation to two types of manure produced in turkey broiler and hen-laying farms. The attempt to assess the applied manure was also considered in relation to the cropping of directly fertilized plants and its subsequent impact in the second year after application.

## 2. Materials and Methods

### 2.1. Characteristics of the Poultry Manure

In the experiment, poultry manures (PMs) from the production of turkey broilers (TB) and egg-laying hens (LH) were used. This natural amendment was obtained from another trial where the effect of Deodoric^®^ mineral–microbial deodorizing preparations (MMDP) on the reduction of ammonia and volatile odorous compounds in poultry houses was studied. MMDP was composed of two parts. The first part was dried material containing a mixture of the following microorganisms: *Lactobacillus plantarum* (ŁOCK 0996), *Leuconostoc mesenteroides* (ŁOCK 0964), *Bacillus megaterium* (ŁOCK 0963), *Bacillus subtilis* (ŁOCK 0962), and *Pseudomonas fluorescens* (ŁOCK 0961). The second part was mineral sorbent composed of perlite (15–20%) and bentonite (80–85%) depending on the birds being bred. More information on MMDP can be found in our publications [28,29].

The experiment was carried out in two series: without the addition of MMDP Deodoric^®^ to poultry manure (PM) (0) and with the addition of MMDP to PM (1).

### 2.2. The Site and Experimental Design

The PMs originated from an experiment carried out in 2016/2017 at the Department of Veterinary Prevention and Feed Hygiene, Faculty of Veterinary Medicine, University of Warmia and Mazury in Olsztyn [28,29]. Manure samples were used for the pot trial in the Experimental Greenhouse of the Faculty of Agriculture and Forestry. The pot experiment was set up in triplicate.

The experiment consisted of two factors: (1) soil amendment, 0-control-without PM, PM from turkey broiler farming without MMDP (TB-0); PM from turkey broiler farming with MMDP (TB-1); PM from egg-laying-hen farming without MMDP (LH-0); PM from egg-laying-hen farming with MMDP (LH-1), mineral fertilizers only (NPK). The second factor (2) was the tested plants: 1. maize LG32 (*Zea maize* L.) for forage (M) (Limagrain Central Europe Société, France) 2. rapeseed forage PVH (*Brassica napus* L. var. Napus x *Brassica rapa* subsp. *Chinensis* (L.) Hanelt) tetraploid hybrid formed as a result of the crossing of winter turnip and Chinese cabbage (P) (KWS SAAT SE, Germany), and 3. sunflower forage (*Helianthus annus* L.) (S) (PLANTA Ltd. Poland). All plants applied in the experiment are used in farming practice as fodder plants for animals.

The experiment was set up in Kick–Brauckmann polyethylene pots, filled with 9 kg of soil sieved through a sieve with a mesh diameter of ø 1 cm. The PM dose corresponded to the rate of 170 kg N ha^−1^, according to the recommendations of the Nitrates Directive (91/676/EEC) [30], which aims to protect the quality of the water against pollution from agricultural sources and to promote the application of good agricultural practices. The manure rate was calculated based on the current N_tot_ content (g kg^−1^ DM) in the manure, the dry matter content (% of DM) in the manure, the weight of the soil in the pot (9 kg pot^−1^), and the planned N rate from the manure ha^−1^ (170 kg N ha^−1^). According to the nitrogen content in PMs in the individual combinations: TB-0, TB-1, LH-0, and LH-1, we used the rates of 49.3, 50.9, 45.6, and 54.9 g of the fresh weight of PM pots^−1^, respectively. In this way, we added exactly 0.64 g per N pot^−1^. The control group, without PMs, was fertilized with 0.64 g N—urea CO(NH_2_)_2_. All experimental pots were fertilized with P and K fertilizers with 0.6 g of P—KH_2_PO_4_, 1.25 g of K—KH_2_PO_4_ (0.75 g K), and K_2_SO_4_ (0.5 g K). During the growth of experimental plants, the soil moisture was kept at the level of 60% of full water capacity.

In the next year, in the same pots, yellow lupine seeds were sown (*Lupinus luteus* L.) as an after-crop plant without any substances added to the soil in order to check the after-effect of the fertilization combinations applied.

### 2.3. Characteristics of the Soil

The soil used in the experiment was taken from the field of the Research Station in Tomaszkowo located near Olsztyn (53°42′35″ N, 20°26′01″ E) from the Ap horizon (0–30 cm). According to the Soil Texture Calculator of Natural Resources Conservation Service of the United States Department of Agriculture (USDA) [31], the soil originated from loamy fine sand. According to FAO/WRB (World Reference Base for Soil Resources) [32], the soil classified as Cambisols—Brown Soils. The basic properties of the soil used in the experiment are presented in Table 1.

### 2.4. Sampling and Chemical Analysis

The chemical analyses of soil, PMs, and plant material samples were carried out in the premises of the Department of Agricultural and Environmental Chemistry laboratory at the University of Warmia and Mazury in Olsztyn.

#### 2.4.1. Soil Analysis

The particle size distribution of the soils used in the experiment was measured in distilled water using a Mastersizer 3000 (Malvern Instruments Limited, Worcestershire, UK) equipped with a Hydro EV module. The soil reaction (pH) was determined in soil/1M solution of KCl suspension, 1:2.5 (*w*/*v*), with a pH SenTix61 electrode and a pH Meter 538 WTW (WTW, Wrocław, Poland). Electrolytic conductivity (EC) was determined in soil/deionized water suspension, 1:5 (*w*/*v*) with a Hanna Instruments HI 8733 Multi–Range PorTable EC Meter (Hanna Instruments, Leighton Buzzard, the United Kingdom). Hydrolytic acidity (HA) was measured according to Kappen’s method after soil extraction with 0.5 M Ca(OAc)_2_ (calcium acetate solution), whose reaction was adjusted to pH 8.2 relative to the soil/Ca(OAc)_2_ solution, 1:2.5 (*w*/*v*) [33]. The sum of base cations (*SBC*) was determined after soil extraction with 1 M NH_4_OAc (ammonium acetate) at pH 7, and calculated as a sum of an individual BC. The cation exchange capacity (*CEC*) is calculated as a sum of *HA* and *SBC* with Formula (1):(1)CEC=SBC+HA

Base saturation (*BS*) is calculated as a percentage of *CEC* occupied by *SBC* with Formula (2):(2)$$BS=\frac{SBC}{CEC}\times 100\%$$

Total carbon content (C_tot_) was assayed on a Shimadzu TOC–L analyzer coupled with an SSM–5000A analyzer for carbon content in solid samples (Shimadzu Corporation, Kyoto, Japan). Total nitrogen (N_tot_) was determined according to Kjeldahl’s distillation method in the soil samples after their wet mineralization in concentrated sulphuric acid using the BUCHI K–355 distillation unit (BÜCHI Labortechnik AG, Flawil, Switzerland). Available phosphorus and potassium were determined using the Egner–Riehm method, and magnesium (Mg) by the Schachtschabel method [34]

The content of individual base cations was determined using flame atomic absorption spectrometry (FAAS) for Mg^2+^, and flame atomic emission spectrometry (FAES) for K^+^, Ca^2+^, and Na^+^. The fast sequential atomic absorption spectrometer VARIAN SpectrAA—FS240 apparatus (Varian Inc., Mulgrave, Australia) was used for the above-mentioned determinations [33].

#### 2.4.2. PM Analysis

The dry weight was determined after 24 h of drying at 65°C with the Binder FED–720 drier (Binder GmbH, Tuttlingen, Germany). Total nitrogen (N_tot_) and total carbon (C_tot_) analyses were carried out using methods such as those used for soil assessment [35]. Phosphorus (P) was calculated by the colorimetric vanadate–molybdate method [33]; magnesium (Mg) with the flame atomic absorption spectrometry (FAAS); and potassium (K), calcium (Ca), and sodium (Na) with the flame atomic emission spectrometry (FAES) using the fast sequential atomic absorption spectrometer VARIAN SpectrAA–FS240 (Varian Inc., Mulgrave, Australia) [33]. The C/N ratio was calculated based on the C_tot_ and N_tot_ content.

#### 2.4.3. Plants Analysis

A leaf’s greenness (SPAD index) is determined as the indicator for the nutritional status of plants [36,37,38,39]. The measurement was performed with the use of the chlorophyll meter SPAD–502Plus (Konica–Minolta, Osaka, Japan) [40]. The harvest of plants was carried out 58 days after sowing with the simultaneous determination of the yield of the aboveground organs.

Yellow lupine-tested plants were harvested in the green pod phase to obtain the aboveground yield.

#### 2.4.4. Experimental Data Analysis

The obtained results were statistically analyzed by performing a two-factor variance analysis ANOVA using a Statistica^®^ v. 13.3 PL software package from TIBCO Software Inc. (Palo Alto, CA, USA) [41], and standard deviation (SD) with Microsoft Excel^®^ for Microsoft 365 MSO (version 2206) (Redmond, WA, USA) [42].

## 3. Results

### 3.1. Properties of PMs

The origin of the poultry manure significantly determined its properties. The manure obtained from laying hens (LH) contained, on average, 15.56% more dry matter than the manure from turkey broilers (TB) (Table 2). The addition of MMDP contributed to the reduction in dry matter content in both TB-1 and LH-1 manure by 2.20 and 3.44%, respectively, to TB-0 and LH-0 manure, without the addition of MMDP.

The PM samples used in the experiment differed significantly in their chemical composition. TB-0 manure contained, on average, much more phosphorus, potassium, magnesium, and sodium compared to LH-0, while LH-0 manure contained more calcium than TB-0 manure. The addition of MMDP increased the phosphorus content in the LH-1 manure, which was not found in the TB-1 manure, but was very abundant in phosphorus. MMDP also increased potassium content—in relation to TB-0 and LH-0—in TB-1 and LH-1 manure by 3.59 and 1.99 g K kg^−1^, respectively, and calcium by 24.5 and 6.12 g Ca kg^−1^, respectively. Under the influence of MMDP, a slight decrease in Na content in the TB-1 manure and a significant increase—by 2.13 g Na kg^−1^ in the case of the LH-1 manure—was found.

The tested PMs differed in nitrogen content, more of which was contained in TB-0 than in LH-0 manure. The addition of MMDP caused an increase in N_tot_ in the case of TB-1 and a significant increase in the case of LH-1 manure. The lower organic carbon content in TB-0 manure compared to LH-0 manure resulted in TB-0 manure having a lower C:N ratio than LH-0 manure. The addition of MMDP contributed to the narrowing of the C:N ratio in the LH-1 manure, which was not found for TB-1 manure.

### 3.2. Reaction, Electrolytic Conductivity, Acidity, and Sorption Complex

Experimental treatments significantly modified the pH of the soil in relation to the pH of the control soil (Table 3).

The highest soil acidification was found after sunflower growth. For this plant, in the control, a drop in pH of 0.55 pH_KCl_ compared to the pH_KCl_ reported in initial soil (Table 1) was shown. To a slightly smaller extent, but also significantly, the reduction in pH_KCl_ was influenced by the growth of maize and rapeseed forage.

The NPK fertilizer application of maize and sunflower, compared to the control series, significantly increased the soil pH_KCl_, which was not shown in the case of rapeseed forage.

Analysis of the soil pH value for the crops showed that this acidifying effect was mitigated by the soil amendments studied.

Electrolytic conductivity (EC) in the soil after plant harvest increased by an average of 1.68 dS m^−1^ in the control compared to the EC recorded in intact soil (Table 3). In general, the lowest EC value comparable to the value found in the control was found in the treatments with NPK fertilizers (2.96 dS m^−1^). However, statistical analysis showed that the response to the applied NPK was crop-dependent. In the case of sunflower, an increase in EC of 0.94 dS m^−1^ was found in relation to the control. In the case of soil after maize harvest, there was a decrease by 1.05 dS m^−1^, while in rapeseed forage, NPK did not significantly change the EC in the soil in comparison with the control.

Soil treatments with PMs had a very pronounced influence on the increase in soil EC. The highest EC value was found in soils amended with TB-0 manure—4.84 dS m^−1^, on average—while the addition of MMDP to the manure reduction in this parameter was noted. On the other hand, LH-0 manure increased the EC value to a lesser extent than TB-0 manure, but the application of MMDP additives, in this case, did not show any changes. LH-1, in relation to the studied crops, had a different effect than LH-0, with EC at the level of 4.17 dS m^−1^.

The mean hydrolytic acidity (HAC) measured in the control soil (Table 4) was increased by an average of 0.69 cmol(^+^) kg^−1^ compared to the intact soil (Table 1) after the harvest of the plants. The plant that increased the HAC value to the greatest extent was sunflower. The acidifying effect was lower for maize and the lowest HAC value was for rapeseed forage. In the case of HAC, significant interactions of fertilizers vs. plants were found. Regarding maize, only the addition of LH-1 had a significant effect on the reduction of the HAC value. In the case of rapeseed forage, this effect was obtained after the use of LH-0, LH-1, and TB-1—in the growth of sunflower, TB-0 manure significantly increased the HAC value, while the remaining PMs did not change the HAC value significantly, and with the control, they constituted a homogeneous group in this respect. Finally, it should be stated that the addition of MMDP to PMs, regardless of the poultry species, had a reducing effect on the soil HAC value, keeping it at a lower level than in the facilities without MMDP additives.

The sum of base cations (SBC) determined after the harvest of the test plants was only slightly increased compared to the control treatment. NPK fertilization contributed to the increase in SBC in relation to the control for each crop under study. The applied PMs constituted an important source of base cations, with LH-0 manure increasing SBC to a greater extent than TB-0. MMDP additions, in both cases, raised the SBC. As a result, the best effects related to the formation of the soil sorption complex were achieved by using LH-1, then TB-1 manure, and LH-0. TB-0 had a similar effect on SBS values as an NPK treatment.

The experimental treatments influenced the size of the sorption complex. All the applied soil amendments and NPK increased the value of the cation exchange capacity (CEC), with NPK acting the weakest in this respect. Under the influence of TB and LH manure, CEC increased by 0.36 and 0.79 cmol(+) kg^−1^, respectively, and the addition of MMDP contributed to a further increase in CEC, to the level, on average, of 7.76 and 8.02 cmol(+) kg^−1^. It should be mentioned that the efficiency of PMs introduced into the soil depended on the crops, but in each case, LH manure provided a more pronounced effect than TB manure, and the addition of MMDP additionally increased this effect, which can also be observed for base saturation (BS).

The BS of the soil used in the experiment was low and amounted to 56.70% (Table 1). The growth of the tested crops reduced the BS for maize, rapeseed forage, and sunflower to 52.91, 53.11, and 51.75%, respectively. Fertilization with NPK of individual plants caused an increase in BS in relation to the control in a significant way, while, as in the case of CEC, the use of PM had a very pronounced impact on the BS evaluated, among which lower values were found after using TB manure than LH. The addition of MMDP to the LH manure increased the BS value to 60.06, 63.48, and 57.94 for LH-1 used in maize, rapeseed forage, and sunflower, respectively.

### 3.3. Total Nitrogen, Total Carbon, and C:N Ratio

The content of total nitrogen (N_tot_) that remained in the soil after crop harvest mainly depended on the use (uptake) of this nutrient by the studied crops. This utilization depended on the applied PMs (Table 5). PMs generally caused a significant increase in N_tot_ content in relation to the control, which was not fertilized and, more importantly, to the series fertilized with NPK. The reported content of N_tot_ after TB-0 and LH-0 application increased, on average, by 0.07 and 0.04 g kg^−1^, respectively, in relation to the control, and by 0.08 and 0.05 kg^−1^ in relation to NPK. When analyzing the interactions of the soil treatments with the studied crops, it should be stated that in the case of maize, no significant effect on the N_tot_ content in the soil under the influence of various PMs was shown. However, it should be noted that all PM treatments increased the total nitrogen content more than NPK did. In the soil analyzed after the harvest of rapeseed forage, a higher N_tot_ residue was manifested for TB-0 and TB-1 than for LH-0 and LH-1, while in the soil after sunflower harvest, more N_tot_ was found in the LH-1 treatment than in TB-0, TB-1, and LH-0. In summary, it can be concluded that a tendency was shown to increase soil abundance in N_tot_ under the influence of PMs, and in TB manures, less N_tot_ remained in the soil after TB-1 application than TB-0, which is likely related to the utilization of nitrogen by crops.

The total carbon (C_tot_) in the soil, on average, amounted to 6.11 g kg^−1^ C_tot_ kg^−1^ (Table 5). The applied NPK fertilizers significantly increased the C_tot_ level by 0.27 g kg^−1^ C_tot_ kg^−1^. A greater increase in C_tot_ concentration was recorded after the use of manures TB-0 and LH-0, by 0.94 and 1.00 g of C_tot_ kg^−1^, respectively. Treatments with the addition of MMDP, i.e., TB-1 and LH-1, were characterized by a slightly lower content of 6.18 and 6.21 g C_tot_ kg^−1^, respectively. Among the tested plants, sunflower left the lowest amount of C_tot_ in the soil, followed by the rapeseed forage; the plant that performed the best in this respect was maize, which left 6.47 g of C_tot_ kg^−1^, on average.

The ratio of carbon to nitrogen is a crucial parameter indicating the current fertility of the soil. Here, this value ranged from 9.39 to 10.26. Both PMs tested caused a significant increase in the C:N ratio in relation to the control, especially in the case of LH-0, where C:N amounted to 10.26. In PM treatment, with the addition of MMDP, lower values of this parameter were recorded. The C:N value was also significantly dependent on the crops grown. In this case, maize had a better effect on soil (C:N = 10.33) than sunflower and rapeseed forage.

### 3.4. Content of Available Forms of Macronutrients

Under the conditions of PM application, an increase in the content of the available fraction of phosphorus (P_av_) in the soil was demonstrated in comparison to the control and NPK treatments (Table 6). The highest content of P_av_ was noted in MMDP treatments, i.e., TB-1 and LH-1, regardless of the grown crops. Higher contents of P_av_. were found in the LH manure treatments than under conditions of TB.

Regarding the available potassium (K_av_) content, it was shown that the NPK and PMs were characterized by a lower content of this nutrient than the control. The highest total K uptake was found in maize, slightly less for sunflowers, and the lowest for rapeseed forage. This plant left most K_av_ in the soil. In this case, no differences were found between the effects of TB-0 and LH-0. The PM treatments, with the addition of MMDP, were characterized by a higher K_av_ content than TB-0 and LH-0.

The concentration of the available magnesium (Mg_av_) was enhanced in all treatments compared to the control. In this respect, in soil with PMs, a higher level of Mg_av_ was found than in the NPK treatment, where an average increase in Mg_av_ in relation to the control was 0.36 mg kg^−1^. Mg_av_ content in the soil was higher after LH application than after TB application, however, the applied MMDP additives resulted in the same Mg_av_ content, which was 4.00 mg kg^−1^, in both the TB-1 and LH-1 treatments. Again, maize uptook more Mg_av_ than sunflower and rapeseed forage.

### 3.5. Plant Properties

#### 3.5.1. Leaf Greenness Index (SPAD)

The test plants used in the research—maize, rapeseed forage, and sunflower—significantly differed in the value of the leaf greenness index (Figure 1). The sunflower leaves (31.31 SPAD units) were the most green, followed by the rapeseed forage leaves (28.02), and the lowest—maize leaves (16.18). When comparing the analyzed PMs, it was found that for each of the tested plants, the manure from TB had a stronger effect than that from LH, as indicated by the means for plants. The effect of TB manure was comparable to that of the NPK used. MMDP showed a slight influence on the SPAD index, a significant increase of which was found only in the case of sunflower from LH manure treatment.

#### 3.5.2. Crop Yield

The aboveground plant yield was determined primarily by the plant species (Figure 2). The highest yield was found for maize, which yielded 546.39 g of aboveground mass pot^−1^. Lower yields were recorded for rapeseed forage and sunflower, respectively: 285.56 and 288.61 g pot^−1^. The highest yield-bearing effect was obtained with the use of NPK 471.67 g pot^−1^ (on average for the tested plants). However, the yield analysis of individual plants showed that NPK significantly increased only the rapeseed forage yield; in the case of other plants, it was shown that in terms of yield, TB manure was comparable to NPK. Lower, but also high yields were obtained in LH manure treatment. It has not been shown that the applied MMDP significantly influenced the yield of the aboveground mass of crops under studies. The effect of the PMs tested on the yielding of the after-crop plant was studied. In the vast majority of experimental treatments, no particular effect of PMs and MMDP additives has been found. However, it was noticed that in the second year, a higher amount of yellow lupine was obtained under LH application conditions than with TB manure in the series after rape forage and sunflower, resulting from a higher nutrient uptake of the test plants in the first year with TB than with LH manure.

## 4. Discussion

Animal production has always been associated with the emission of harmful compounds into the atmosphere. They are burdensome substances for people living in agricultural areas and working in livestock facilities [43,44]. They contribute to chronic headaches, irritability, and stress in livestock workers [19,28,29,45]. Odorous substances also affect the health of livestock and reduce the efficiency of animal production [28,29,46]. In light of the changing regulations, more attention is focused on maintaining proper animal welfare [47,48]. The air inside livestock buildings, especially with a high density of livestock, is characterized not only by a high content of odorous compounds (a cocktail of more than 168 compounds) [45], but also a high concentration of mainly ammonia, hydrogen sulfide, nitrogen oxides, and dust of various origins [44]. A higher mortality rate of animals related to impaired respiratory systems was observed in such facilities [19,49]. In the case of poultry, the harmful effect of the factors contributes to a reduction in the productivity of egg-laying hens [28], and in the case of turkey broilers, a reduction in weight gain [29]. Worldwide efforts to reduce harmful gas emissions from manure generated in livestock facilities have already shown promising results [16,19]. The use of various preparations—in our case, MMDP Deodoric^®^—reduced the nuisance of harmful gases present in livestock facilities. After a series of tests, it was shown that the preparation had a beneficial effect in reducing harmfulness, especially ammonia in relation to turkey broilers and egg-laying-hen houses.

At the same time, a question arose as to whether, and if so, how, the applied preparations affect the waste, i.e., manure, obtained during the breeding process? What is the “added value” of using MMDP Deodoric^®^?

The results obtained in our trial showed some effect of PM application on soil quality indicators. In general, PM treatment resulted in an increase in soil pH and a decrease in HAC in relation to the soil from the control and NPK treatments. This was due to the high Ca content in PMs. Adekiya et al. [11] also observed an increase in the pH value after the application of PM [11]. A higher increase in pH in our investigation was observed with the use of LH than with TB manure. Poultry manure is characterized by the highest calcium content among the manures obtained from farm animals [18,50]. Here, the Ca concentration in the manure of TB and LH was remarkably high, on average, 44.77 and 99.80 g Ca kg^−1^, respectively, which resulted from the experimental nature of poultry farming. This content in PMs under production conditions is, on average, 16.2–24.0 g Ca kg^−1^, while the Ca content in cow and pig manure amounts only to 4.2 and 4.4 g Ca kg^−1^, respectively [17,50]. The high content of Ca in the analyzed PMs results from the high concentration of calcium in poultry diets [51,52]. The use of PMs in our research increased the salinity of the soil. PMs, due to the high content of ammonia and the presence of other water-soluble salts, can cause these types of changes in the soil [53]. Generally, organic manures mitigate the salinity stress caused by mineral fertilization, as reported by Dhaliwal et al. [54]. The authors indicated that the plots amended with organic manures along with mineral fertilizers were characterized by lower EC values than the plots with mineral fertilizers. In our investigation, the opposite was observed. However, the addition of MMDP to the manure mitigated the increase in EC. One of the important parameters to consider when assessing soil fertility and its nutrient retention capacity is indicated by CEC. In the presented study, all the PMs used significantly increased soil CEC. Similar observations were made by Zolfi-Bavariani et al. [53], who noted a significant increase in CEC after the application of PM. The authors showed a positive correlation between CEC and soil pH [53], which could be the result of a large amount of alkaline cations introduced with PMs [4]. The use of MMDP Deodoric^®^ brought an additional benefit in relation to SBC and CEC, which was the effect of MMDP components used in the production, such as perlite containing oxides of silicon, aluminum, sodium, potassium, iron, magnesium, and calcium [21] and clay minerals such as bentonite, containing oxides of silicon and iron, as well as Mg, K and Ca [24]. The research by Czaban and Siebielec [55] has shown that soils with the addition of bentonite were characterized by a higher pH and content of available Mg than the control soil. The soil with the addition of bentonite contained more Ca, Mg, Zn, and Mn in total and absorbable P and K than the other soils. Moreover, it had a higher CEC. According to Czaban et al. [22], the soils with the addition of bentonite contained significantly higher amounts of organic carbon and total nitrogen than the control. As a result of the research conducted, a significant, beneficial effect of PMs on soil BS was demonstrated, especially PMs with the addition of MMDP Deodoric^®^.

PMs tended to be higher in most nutrients, break down faster, and have a higher value compared to manures from other animal species, e.g., cows, horses, or pigs. This is evident, for example, in the case of N_tot_—the content in horse manure is 10.5 g of N_tot_ kg^−1^ of DM, while in PMs, this content is, on average, 32.8 g of N_tot_ kg^−1^ of DM [51]. However, the manure analyzed by us was characterized by a relatively low content—12.53–13.97 g N_tot_ kg^−1^ of DM on average—while PMs usually contain 30–50 g N_tot_ kg^−1^ of DM [18,50,51]. In our case, this was due to the addition of more straw to the litter, which was expected to reduce the moisture of the litter. The dry matter content of the analyzed PMs was 31.22 and 46.78% for TB, and LH, respectively; while fresh and humid litter contain usually 12–20% of DM, a litter stored for a period of one month contains about 20–27% of DM [56]. This higher proportion of straw, with a short period of manure maturation, resulted in an increase in C_tot_ content and a significant extension of the C:N ratio in PMs compared to the data found in the literature. In the reported studies, the average C:N value amounted to 24.32, while according to the literature, it ranged from 6 to 7 [6,57] or from 3 to 10 [56], but a range from 1 to 27 has been also reported [58]. Manure in the Agbede study [59] was characterized by higher humidity, lower C_tot_ content, and higher N_tot_, which, with a 3-week maturation time, resulted in the C:N ratio being set at the level of 7.8. Too low N content or too high C content in manure always results in N blocking by microorganisms decomposing the carbon-rich matrix of the litter, which causes the biological sorption of soil nitrogen reserves from the soil. However, the high C content in the manure contributes to the higher composting efficiency. In the studies of Liu et al. [60], the C:N value of 17.3 significantly enhanced the composting efficiency to the C:N ratio of 9.61 [60]. According to other reports, the rate of 10 Mg ha^−1^ PM is comparable to the rate of 300 kg ha^−1^ of 15 N-15 P-15 K fertilizer [6], as manure contains approximately 3–5% of nitrogen, 0.9–3.5% of phosphorus, and 1.5–3% of potassium [56]. The amendment with poultry manure increased not only the content of organic carbon [2,8] and organic matter in the soil [1,4,5,8], but also N_tot_ [4,5], exchangeable K, Mg, Ca [4], P [4,5,8], and K [8]. As can be seen from the above, the use of PMs in agriculture undeniably affected the properties of the soil, and in consequence, the productivity of plants. Herein, it was shown that PMs, with the addition of MMDP Deodoric^®^ used as a litter additive, increased the pool of available nitrogen in the soil, which reflected the yields of the obtained plants. Particular beneficial effects of MMDP have been manifested with LH manure used for maize. The applied organic amendments naturally contributed to the improvement in the soil supply, with the available P, K, and Mg forms of what others also pointed out [4,11].

The above-described effects of PMs were reflected in the development of the leaf greenness index. The leaf greenness index (SPAD) is a reliable indicator of the nutritional status of plants. In so-called precise agriculture, it is used in particular to determine the crop demand for nitrogen, which is directly correlated with the plant yield [36,37,38,39]. The SPAD values determined in the presented studies showed that TB manure application was comparable to NPK fertilization. The use of MMDP consistently resulted in an increase in the SPAD index in relation to LH-0 and TB-0 treatments.

The literature emphasizes the positive impact of the use of poultry manure on the growth and yield of basic crops, including maize plants (*Zea mays*) [1,9,10,11], carrot (*Daucus carota*) [6], coconuts (*Cocos nucifera*) [7], the biomass of grass (*Panicum maximum*) [8], yam (*Dioscorea rotundata*) [4], and tomatoes (*Solanum lycopersicum*) [5]. In addition, as a result of the application of PMs, an increase in the dry matter content of plants and a considerable enhancement in concentration of N, P and K, Ca, and Mg in plant tissues are also observed [2,4,5]. The application of PM balanced the rate in terms of the amount of nitrogen introduced, with a rate of 170 kg N ha^−1^, thus, resulting in a different level of yields obtained, depending on the crops. In the first year of the study, TB manure was more yield-bearing than the LH one. The applied PM rates in the case of sunflower gave yields comparable to that of NPK, as mentioned by Agbede [6,59]. In the case of rapeseed forage, NPK was more beneficial than TB and LH manures in terms of the obtained yield. In the case of maize, TB manure was comparable to NPK, while in LH manure treatment, a slightly lower yield was noted. The effect that was obtained in the first year of the study, resulting from the applied PMs and NPK, in the second year, basically disappeared; however, the analysis of the averages showed a continued beneficial effect of LH manure on the yield of the studied crops The applied MMDP Deodoric^®^ additive slightly increased the crops’ yield in the first year—such an effect was not shown in the case of lupine, which was grown as an after-crop. In such a way, it can be explained that the positive yield-bearing effect of the applied manure under study was temporary. Furthermore, nitrogen, which was present in organic amendments, significantly affected yields in the first year, whereas lupine grown in the second year due to microbial symbiosis did not respond to the soil nitrogen level.

## 5. Conclusions

The PMs analyzed in our research differed significantly in terms of dry matter content and chemical composition, which was determined by their origin. PMs from egg-laying hens contained more DM and considerably more calcium than PMs from turkey broilers—this was associated with a higher content of calcium in the LH diet, while TB manure was characterized by a higher content of potassium phosphorus, magnesium, and sodium, as well as carbon. In terms of the N_tot_ content of TB manure, it did not differ significantly from LH manure. The addition of MMDP, by increasing the absorbing capacity of the litter and due to the activity of microorganisms, resulted in a higher binding ability of nutrients such as nitrogen, potassium, calcium, as well as carbon in LH manure, and additionally phosphorus in TB manure, in relation to the treatments where MMDP was not applied.

Due to the presence of alkaline ions, the application of PMs resulted in an increase in pH and a decrease in HAC. As a negative effect, an increase in EC was also observed, however, it was counteracted by the addition of MMDP. MMDP improved soil properties to a greater extent than the yields of maize, rapeseed forage, and sunflower. Regarding the soil, MMDP had a positive effect on the sorption complex, especially on values of SBC, CEC, and BS. Regarding plants, our research showed that PM with MMDP slightly increased the SPAD index of leaves and improved the yield. The main benefit associated with the use of MMDP in poultry production was the reduction in harmful gases and dust emissions to the atmosphere, while the ‘added value’ is the improvement in the chemical and physical properties of PMs, which had a positive impact on important soil parameters.

## Figures and Tables

**Figure 1 ijerph-19-16639-f001:**
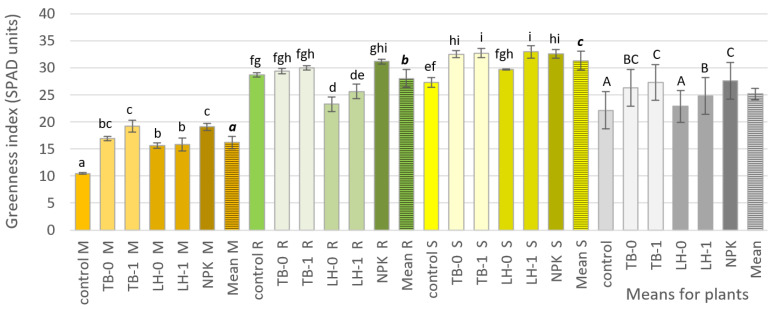
Leaf greenness index (SPAD) of maize, rapeseed forage, and sunflower under different PM treatments. Values are means, error bars indicate the standard error (SE) of the means; means followed by different letters are significantly different by the LSD*_p_*_≤0.05_ test (uppercase—the differences between fertilizers used, lowercase italic—the differences between plant reactions, lowercase regular—interaction of fertilization vs. plant−soil reactions); M—Maize, R—Rapeseed forage S—Sunflower; *n* = 3.

**Figure 2 ijerph-19-16639-f002:**
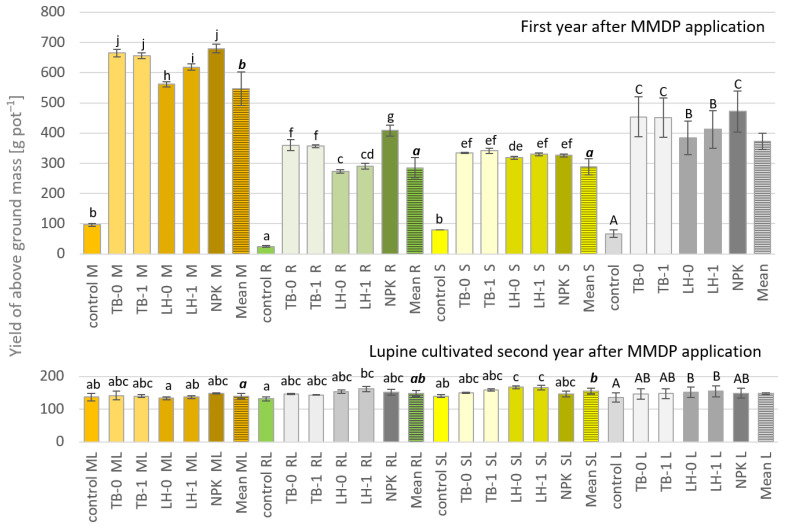
The aboveground yield of tested plants first year and second year after PM application. Values are means, error bars indicate the standard error (SE) of the means; means followed by the LSD*_p_*_≤0.05_ test (uppercase—the differences between fertilizers used, lowercase italic—the differences between plant reactions, lowercase regular—the interaction fertilization vs. the plant−soil reaction); M−Maize, R—Rapeseed forage S—Sunflower, ML—Maize-Lupine, RL—Rapeseed Forage-Lupine, SL−Sunflower-Lupine; *n* = 3.

**Table 1 ijerph-19-16639-t001:** Characteristics of the soil used in the experiment.

Parameter	Unit	Value
Particle size distribution: Sand 0.05–2 mm	%	75.10
Silt 0.002–0.05 mm	%	24.30
Clay ≤ 0.002 mm	%	0.60
Soil texture	–	loamy fine sand
Soil reaction (pH_KCl_)	–log_10_[H^+^]	4.82
Electrolytic conductivity (EC)	dS m^−1^	1.20
Hydrolytic acidity (HAC)	cmol(+) kg^−1^	2.68
Sum of base cations (SBC)	cmol(+) kg^−1^	3.51
Cation exchange capacity (CEC)	cmol(+) kg^−1^	6.19
Base saturation (BS)	%	56.70
Total nitrogen (N_tot_)	g kg^−1^	0.63
Total carbon (C_tot_)	g kg^−1^	4.52
C/N	ratio	7.17

**Table 2 ijerph-19-16639-t002:** Chemical characteristics of poultry manures (PMs).

Parameter	Value					
	TB-0	TB-1	Mean	LH-0	LH-1	Mean
Dry matter (DM) (%)	32.32 ± 0.158 ^b^	30.12 ± 0.069 ^a^	31.22 ^A^	48.50 ± 0.464 ^d^	45.06 ± 0.271 ^c^	46.78 ^B^
Total nitrogen (N_tot_) (g kg DM^−1^)	12.53 ± 0.045 ^b^	12.94 ± 0.029 ^c^	12.74 ^A^	11.62 ± 0.036 ^a^	13.97 ± 0.091 ^d^	12.80 ^A^
Phosphorus (P) (g kg DM^−1^)	124.02 ± 1.129 ^a^	122.37 ± 1.478 ^a^	123.19 ^B^	80.74 ± 0.462 ^b^	89.77 ± 1.673 ^c^	85.26 ^A^
Potassium (K) (g kg DM^−1^)	25.00 ^±^ 0.177 ^c^	28.59 ± 0.124 ^d^	26.80 ^B^	20.54 ± 0.088 ^b^	22.53 ± 0.082 ^a^	21.54 ^A^
Calcium (Ca) (g kg DM^−1^)	32.52 ± 0.419 ^a^	57.02 ± 0.586 ^b^	44.77 ^A^	96.74 ± 0.772 ^c^	102.86 ± 0.406 ^d^	99.80 ^B^
Magnesium (Mg) (g kg DM^−1^)	7.62 ± 0.051 ^c^	7.26 ± 0.001 ^b^	7.44 ^B^	5.02 ± 0.051 ^a^	7.15 ± 0.011 ^b^	6.09 ^A^
Sodium (Na) (g kg DM^−1^)	6.49 ± 0.206 ^b^	6.48 ± 0.019 ^b^	6.49 ^B^	5.39 ± 0.085 ^a^	6.42 ± 0.076 ^b^	5.90 ^A^
Total carbon (C_tot_) (g kg DM^−1^)	318.45 ± 0.636 ^b^	319.45 ± 1.438 ^b^	318.95 ^B^	319.60 ± 2.546 ^b^	295.25 ± 2.192 ^a^	307.43 ^A^
C/N (ratio)	25.63 ± 0.110 ^b^	24.80 ± 0.094 ^b^	25.05 ^B^	27.42 ± 0.300 ^c^	20.85 ± 0.141 ^a^	24.32 ^A^

Values are means; ± standard error of means; means followed by different letters are significantly different by the LSD*_p_*_≤0.05_ test (uppercase regular—the differences between TB and LH manures; lowercase—the interaction MMDP × PMs); *n* = 3.

**Table 3 ijerph-19-16639-t003:** Effect of PM application on the soil reaction and electrical conductivity of the soil after the first year of plant harvest.

Treatment	Maize	Rapeseed Forage	Sunflower	Mean
pH 1M KCl (−log_10_H^+^)				
Control	4.31 ± 0.012 ^ab^	4.37 ± 0.010 ^c^	4.27 ± 0.012 ^a^	4.32 ± 0.030 ^A^
TB-0	4.53 ± 0.012 ^e^	4.82 ± 0.007 ^i^	4.41 ± 0.015 ^d^	4.59 ± 0.061 ^D^
TB-1	4.61 ± 0.019 ^g^	4.66 ± 0.003 ^h^	4.43 ± 0.012 ^d^	4.56 ± 0.035 ^C^
LH-0	4.54 ± 0.003 ^e^	4.81 ± 0.000 ^i^	4.54 ± 0.009 ^e^	4.63 ± 0.045 ^E^
LH-1	4.98 ± 0.038 ^j^	5.03 ± 0.003 ^k^	4.56 ± 0.018 ^ef^	4.86 ± 0.075 ^F^
NPK	4.59 ± 0.003 ^fg^	4.33 ± 0.003 ^b^	4.33 ± 0.003 ^bc^	4.42 ± 0.043 ^B^
Mean	4.59 ± 0.049 *^B^*	4.67 ± 0.061 *^C^*	4.42 ± 0.025 *^A^*	4.56 ± 0.030
Electrolytic conductivity (EC) (dS m^−1^)				
Control	3.41 ± 0.093 ^de^	2.61 ± 0.049 ^ab^	2.63 ± 0.019 ^ab^	2.88 ± 0.134 ^A^
TB-0	5.26 ± 0.257 ^k^	5.25 ± 0.127 ^k^	4.00 ± 0.061 ^fg^	4.84 ± 0.227 ^E^
TB-1	4.73 ± 0.121 ^ij^	4.96 ± 0.186 ^jk^	3.52 ± 0.060 ^e^	4.40 ± 0.233 ^D^
LH-0	3.74 ± 0.012 ^ef^	3.44 ± 0.124 ^de^	3.11 ± 0.100 ^cd^	3.43 ± 0.102 ^B^
LH-1	4.30 ± 0.226 ^gh^	4.51 ± 0.116 ^hi^	3.71 ± 0.086 ^ef^	4.17 ± 0.143 ^C^
NPK	2.36± 0.070 ^a^	2.93 ± 0.034 ^bc^	3.57 ± 0.023 ^e^	2.96 ± 0.177 ^A^
Mean	3.97 ± 0.234 *^B^*	3.95 ± 0.248 *^B^*	3.42 ± 0.109 *^A^*	3.78 ± 0.122

Values are means; ± standard error of means; means followed by different letters are significantly different by the LSD*_p_*_≤0.05_ test (uppercase regular—the differences between soil treatments used; uppercase italic—differences between plant reactions; lowercase—the interaction of soil treatments used vs. the plant−soil reaction); *n* = 3.

**Table 4 ijerph-19-16639-t004:** The effect of PM application on the properties of the soil sorption complex after the first year of plant harvest.

Treatment	Maize	Rapeseed Forage	Sunflower	Mean
Hydrolytic Acidity (HAC) (cmol(^+^) kg^−1^)				
Control	3.49 ± 0.022 ^jk^	3.24 ± 0.035 ^de^	3.38 ± 0.000 ^gh^	3.37 ± 0.026 ^C^
TB-0	3.47 ± 0.009 ^ijk^	3.29 ± 0.009 ^ef^	3.53 ± 0.043 ^k^	3.43 ± 0.038 ^D^
TB-1	3.45 ± 0.043 ^hij^	3.08 ± 0.026 ^c^	3.41 ± 0.022 ^hij^	3.31 ± 0.062 ^B^
LH-0	3.41 ± 0.022 ^ghij^	3.00 ± 0.000 ^b^	3.45 ± 0.000 ^hijk^	3.29 ± 0.072 ^B^
LH-1	3.23 ± 0.000 ^de^	2.89 ± 0.022 ^a^	3.40 ± 0.013 ^ghi^	3.17 ± 0.075 ^A^
NPK	3.18 ± 0.000 ^d^	3.00 ± 0.043 ^b^	3.34 ± 0.022 ^fg^	3.17 ± 0.051 ^A^
Mean	3.37 ± 0.030 *^B^*	3.08 ± 0.035 *^A^*	3.42 ± 0.016 *^C^*	3.29 ± 0.026
Sum of base cations (SBC) (cmol(+) kg^−1^)				
Control	3.92 ± 0.069 ^b^	3.67 ± 0.040 ^a^	3.62 ± 0.012 ^a^	3.74 ± 0.057 ^A^
TB-0	4.05 ± 0.029 ^c^	3.88 ± 0.046 ^b^	4.18 ± 0.012 ^d^	4.04 ± 0.046 ^B^
TB-1	4.40 ± 0.000 ^e^	4.65 ± 0.029 ^fg^	4.28 ± 0.069 ^de^	4.44 ± 0.059 ^C^
LH-0	4.38 ± 0.012 ^e^	4.87 ± 0.040 ^h^	4.55 ± 0.087 ^f^	4.60 ± 0.077 ^D^
LH-1	4.85 ± 0.029 ^h^	5.02 ± 0.012 ^i^	4.68 ± 0.046 ^g^	4.85 ± 0.052 ^E^
NPK	4.25 ± 0.029 ^d^	4.05 ± 0.029 ^c^	3.80 ± 0.058 ^b^	4.03 ± 0.068 ^B^
Mean	4.31 ± 0.073 *^B^*	4.36 ± 0.125 *^B^*	4.19 ± 0.093 *^A^*	4.28 ± 0.057
Cation exchange capacity (CEC) (cmol(+) kg^−1^)				
Control	7.41 ± 0.091 ^d^	6.91 ± 0.075 ^a^	7.00 ± 0.012 ^ab^	7.10 ± 0.052 ^A^
TB-0	7.52 ± 0.038 ^d^	7.17 ± 0.038 ^c^	7.71 ± 0.032 ^a^	7.46 ± 0.081 ^C^
TB-1	7.85 ± 0.043 ^efgh^	7.73 ± 0.055 ^ef^	7.69 ± 0.048 ^e^	7.76 ± 0.034 ^D^
LH-0	7.79 ± 0.033 ^efg^	7.87 ± 0.040 ^fgh^	8.00 ± 0.087 ^hi^	7.89 ± 0.042 ^E^
LH-1	8.08 ± 0.029 ^i^	7.91 ± 0.010 ^gh^	8.08 ± 0.059 ^i^	8.02 ± 0.034 ^F^
NPK	7.43 ± 0.029 ^d^	7.05 ± 0.072 ^abc^	7.14 ± 0.051 ^bc^	7.21 ± 0.063 ^B^
Mean	7.68 ± 0.062 *^C^*	7.44 ± 0.100 *^A^*	7.60 ± 0.100 *^B^*	7.57 ± 0.052
Base saturation (BS) (%)				
Control	52.91 ± 0.29 ^b^	53.11 ± 0.01 ^bc^	51.75 ± 0.08 ^a^	52.59 ± 0.44 ^A^
TB-0	53.89 ± 0.12 ^cd^	54.15 ± 0.36 ^d^	54.25 ± 0.37 ^d^	54.10 ± 0.16 ^B^
TB-1	56.05 ± 0.31 ^ef^	60.19 ± 0.05 ^i^	55.63 ± 0.56 ^e^	57.29 ± 0.75 ^D^
LH-0	56.21 ± 0.09 ^ef^	61.88 ± 0.20 ^j^	56.86 ± 0.47 ^fg^	58.32 ± 0.91 ^E^
LH-1	60.06 ± 0.14 ^i^	63.48 ± 0.23 ^k^	57.94 ± 0.15 ^h^	60.49 ± 0.81 ^F^
NPK	57.20 ± 0.17 ^gh^	57.45 ± 0.18 ^gh^	53.23 ± 0.48 ^bc^	55.96 ± 0.70 ^C^
Mean	56.05 ± 0.56 *^B^*	58.38 ± 0.93 *^C^*	54.95 ± 0.53 *^A^*	56.46 ± 0.44

Values are means; ± standard error of means; means followed by different letters are significantly different by the LSD*_p_*_≤0.05_ test (uppercase regular—the differences between treatments, uppercase italic—the differences between plant reactions, lowercase—interaction treatments vs. the plant−soil reaction); *n* = 3.

**Table 5 ijerph-19-16639-t005:** The effect of PM application on the content of total nitrogen, total carbon, and C:N ratio in the soil after the first year of plant harvest.

Treatment	Maize	Rapeseed Forage	Sunflower	Mean
Total Nitrogen (N_tot_) (g kg^−1^)				
Control	0.57 ± 0.003 ^a^	0.62 ± 0.008 ^cde^	0.57 ± 0.012 ^a^	0.59 ± 0.005 ^A^
TB-0	0.66 ± 0.004 ^fgh^	0.67 ± 0.002 ^h^	0.65 ± 0.010 ^defgh^	0.66 ± 0.004 ^C^
TB-1	0.63 ± 0.010 ^cdefg^	0.66 ± 0.006 ^gh^	0.63 ± 0.004 ^cdef^	0.64 ± 0.006 ^B^
LH-0	0.65 ± 0.006 ^fgh^	0.62 ± 0.006 ^cd^	0.63 ± 0.012 ^cdef^	0.63 ± 0.007 ^B^
LH-1	0.65 ± 0.001 ^efgh^	0.64 ± 0.008 ^cdefg^	0.61 ± 0.014 ^bc^	0.63 ± 0.007 ^B^
NPK	0.58 ± 0.006 ^a^	0.59 ± 0.008 ^ab^	0.57 ± 0.016 ^a^	0.58 ± 0.006 ^A^
Mean	0.62 ± 0.009 *^B^*	0.63 ± 0.007 *^B^*	0.61 ± 0.008 *^A^*	0.62 ± 0.005
Total Carbon (C_tot_) (g kg^−1^)				
Control	5.47 ± 0.12 ^a^	5.54 ± 0.02 ^abc^	5.51 ± 0.04 ^ab^	5.51 ± 0.07 ^A^
TB-0	7.07 ± 0.16 ^h^	6.38 ± 0.20 ^fg^	5.89 ± 0.10 ^cde^	6.45 ± 0.19 ^D^
TB-1	6.58 ± 0.10 ^g^	6.13 ± 0.16 ^ef^	5.84 ± 0.01 ^bcde^	6.18 ± 0.12 ^C^
LH-0	7.14 ± 0.22 ^h^	6.50 ± 0.06 ^g^	5.90 ± 0.06 ^cde^	6.51 ± 0.19 ^D^
LH-1	6.67 ± 0.01 ^g^	6.09 ± 0.10 ^def^	5.86 ± 0.14 ^bcde^	6.21 ± 0.13 ^C^
NPK	5.88 ± 0.08 ^bcde^	5.71 ± 0.10 ^abc^	5.74 ± 0.09 ^abcd^	5.78 ± 0.05 ^B^
Mean	6.47 ± 0.15 *^C^*	6.06 ± 0.09 *^B^*	5.79 ± 0.04 *^A^*	6.11 ± 0.07
C:N (ratio)				
Control	9.64 ± 0.22 ^bcd^	8.90 ± 0.08 ^a^	9.61 ± 0.27 ^bcd^	9.39 ± 0.08 ^A^
TB-0	10.74 ± 0.18 ^gh^	9.54 ± 0.27 ^bcd^	9.11 ± 0.20 ^ab^	9.80 ± 0.27 ^BC^
TB-1	10.39 ± 0.20 ^fgh^	9.26 ± 0.17 ^abc^	9.27 ± 0.04 ^abc^	9.64 ± 0.20 ^AB^
LH-0	10.91 ± 0.24 ^h^	10.49 ± 0.12 ^fgh^	9.37 ± 0.23 ^abc^	10.26 ± 0.25 ^D^
LH-1	10.24 ± 0.01 ^efg^	9.57 ± 0.03 ^bcd^	9.57 ± 0.03 ^bcd^	9.79 ± 0.11 ^BC^
NPK	10.06 ± 0.03 ^def^	9.73 ± 0.31 ^cde^	10.13 ± 0.13 ^def^	9.97 ± 0.12 ^CD^
Mean	10.33 ± 0.12 *^B^*	9.58 ± 0.14 *^A^*	9.51 ± 0.10 *^A^*	9.81 ± 0.08

Values are means; ± standard error of means; means followed by different letters are significantly different by the LSD*_p_*_≤0.05_ test (uppercase regular—the differences between treatments, uppercase italic—the differences between plants reaction, lowercase—the interaction treatments vs. the plant−soil reaction); *n* = 3.

**Table 6 ijerph-19-16639-t006:** The effect of PM application on the content of available forms of macronutrients in the soil after the first year of plant harvest.

Treatment	Maize	Rapeseed Forage	Sunflower	Mean
Available phosphorus (P_av_) (mg kg^−1^)				
Control	9.16 ± 0.13 ^a^	9.41 ± 0.05 ^a^	9.23 ± 0.07 ^a^	9.27 ± 0.15 ^A^
TB-0	11.47 ± 0.03 ^cd^	12.31 ± 0.19 ^ghi^	11.61 ± 0.25 ^cde^	11.79 ± 0.16 ^C^
TB-1	11.81 ± 0.10 ^de^	12.62 ± 0.23 ^h^	11.84 ± 0.01 ^def^	12.09 ± 0.15 ^D^
LH-0	10.74 ± 0.02 ^b^	12.05 ± 0.03 ^efg^	11.31 ± 0.05 ^c^	11.36 ± 0.19 ^B^
LH-1	12.31 ± 0.01 ^gh^	13.15 ± 0.10 ^i^	11.33 ± 0.12 ^c^	12.26 ± 0.27 ^D^
NPK	12.03 ± 0.00 ^efg^	12.05 ± 0.41 ^efg^	11.28 ± 0.06 ^c^	11.78 ± 0.17 ^C^
Mean	11.25 ± 0.26 *^A^*	11.93 ± 0.30 *^B^*	11.10 ± 0.21 *^A^*	11.42 ± 0.15
Available potassium (K_av_) (mg kg^−1^)				
Control	8.73 ± 0.01 ^fg^	11.04 ± 0.13 ^i^	9.31 ± 0.09 ^gh^	9.69 ± 0.27 ^D^
TB-0	5.23 ± 0.02 ^b^	8.66 ± 0.05 ^f^	5.70 ± 0.14 ^bc^	6.53 ± 0.54 ^A^
TB-1	6.78 ± 0.02 ^e^	8.76 ± 0.12 ^fg^	6.31 ± 0.09 ^cde^	7.28 ± 0.38 ^B^
LH-0	4.34 ± 0.41 ^a^	9.50 ± 0.21 ^h^	6.20 ± 0.01 ^cde^	6.68 ± 0.77 ^A^
LH-1	5.81 ± 0.26 ^bcd^	10.94 ± 0.14 ^i^	6.35 ± 0.27 ^de^	7.70 ± 0.82 ^C^
NPK	5.85 ± 0.13 ^cd^	8.46 ± 0.44 ^f^	8.23 ± 0.22 ^f^	7.52 ± 0.44 ^BC^
Mean	6.12 ± 0.34 *^A^*	9.56 ± 0.27 *^C^*	7.02 ± 0.32 *^B^*	7.57 ± 0.27
Available magnesium (Mg_av_) (mg kg^−1^)				
Control	3.18 ± 0.05 ^a^	3.18 ± 0.05 ^ab^	3.49 ± 0.03 ^cd^	3.28 ± 0.04 ^A^
TB-0	3.45 ± 0.03 ^c^	3.88 ± 0.19 ^efg^	3.85 ± 0.09 ^efg^	3.73 ± 0.09 ^B^
TB-1	3.73 ± 0.14 ^e^	4.28 ± 0.01 ^h^	3.99 ± 0.07 ^fg^	4.00 ± 0.09 ^D^
LH-0	3.73 ± 0.01 ^e^	4.07 ± 0.11 ^gh^	3.78 ± 0.06 ^ef^	3.86 ± 0.06 ^C^
LH-1	3.90 ± 0.07 ^efg^	4.10 ± 0.09 ^gh^	3.99 ± 0.03 ^fg^	4.00 ± 0.04 ^D^
NPK	3.81 ± 0.01 ^ef^	3.71 ± 0.05 ^de^	3.41 ± 0.03 ^bc^	3.64 ± 0.06 ^B^
Mean	3.63 ± 0.06 *^A^*	3.87 ± 0.09 *^C^*	3.75 ± 0.06 *^B^*	3.75 ± 0.04

Values are means; ± standard error of means; means followed by different letters are significantly different by the LSD*_p_*_≤0.05_ test (uppercase regular—the differences between treatments, uppercase italic—the differences between plant reactions, lowercase—the interaction treatments vs. the plant−soil reaction); *n* = 3.

## Data Availability

Not applicable.

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
