# Peer review of "Effect of Mineral–Microbial Deodorizing Preparation on the Value of Poultry Manure as Soil Amendment"

_ijerph, 2022, doi:10.3390/ijerph192416639_

Round 1

Reviewer 1 Report

The manuscript entitled “Effect of mineral-microbial deodorizing preparation on the value of poultry manure as soil amendment” by Zolnowski et al presents the fertilizing effects of two sources of poultry manure treated with and without mineral-microbial deodorizing preparations (MMDP). Three crops were studied, namely maize, rapeseed forage, and sunflower, and important soil properties were analyzed after harvest. The treatment with MMDP tend to have some beneficial effects on both plant and soil properties, in addition to increased hygienic circumstances for poultry growth. The manuscript is very well written and structured and the authors clearly explain the results that they have obtained. The manuscript is worth publishing after addressing the following small issues:

1. Some corrections should be made to the abstract: 

Line 12:  “do not-poses” should be changed to “do not pose”

Line 14: “C’H4” should be changed to “CH4”

Line 18: “the litter of: turkey broilers and eggs-lying hens (LH), (TB),respectively” should be changed to “the litter of turkey broilers (TB) and eggs-lying hens (LH)”

2. In the abstract and in the conclusions the authors should briefly mention the results obtained from the second year study with lupine.

3. Abbreviations such as PM and also that of chemical elements should be used consistently throughout the manuscript. Give the full name at first use and introduce the abbreviation, and then use the abbreviation in all subsequent mentions.

4. Lines 60-63: “MMDP is composed also with the carriers of microorganisms usually perlite or bentonite [19], which, apart from fulfilling this function, also considerably increase the absorption capacity of these preparations to the liquid components of manure.” This sentence is confusing, please rephrase.

5. Line 91-92: what do the authors mean with “amount of crops”? Please clarify.

6. Line 115: “In the spring 2017 was transported…”. What was transported? Please specify.

7. The use of commas and dots for decimals should be used consistently throughout the manuscript. For example, line 138 contains both “0.64” and “0,64”, and also Table 2 contains a comma. Please check and correct throughout the manuscript.

8. Table 2: the column headings of this table are not clear. Please use the previously introduced abbreviations (TB-0, TB-1, LH-0 and LH-1) as column headings.

9. Table 2: are the contents of Ntot, P, K, Ca, Mg, Na, and TC expressed per fresh weight or per dry weight? Please specify.

10. Please check the consistent style of the tables. In tables 2-5, part of the first row is underlined and sometimes in bold.

11. In the manuscript, sometimes total carbon is expressed by TC (e.g., in section 2.4) and sometimes by Ctot (e.g., in section 3.3). Is there a difference between the two?

12. The authors show that the application of MMDP generally improves the soil properties such as SBC, CEC, and BS. However, in the second year study this did not translate in a higher yield for lupine. Why not? Please clarify.

Author Response

Dear Reviewer,

Thank you very much for your detailed and competent review.

Please find my answers point by point below

  1. Some corrections should be made to the abstract: 

Line 12:  “do not-poses” should be changed to “do not pose”

It was changed

Line 14: “C’H4” should be changed to “CH4”

Done – thank you

Line 18: “the litter of: turkey broilers and eggs-lying hens (LH), (TB),respectively” should be changed to “the litter of turkey broilers (TB) and eggs-lying hens (LH)”

Done – thank you

  1. In the abstract and in the conclusions the authors should briefly mention the results obtained from the second year study with lupine.

Thank you - please see line 24

  1. Abbreviations such as PM and also that of chemical elements should be used consistently throughout the manuscript. Give the full name at first use and introduce the abbreviation, and then use the abbreviation in all subsequent mentions.

This remark was taken into consideration and corrections were made.

  1. Lines 60-63: “MMDP is composed also with the carriers of microorganisms usually perlite or bentonite [19], which, apart from fulfilling this function, also considerably increase the absorption capacity of these preparations to the liquid components of manure.” This sentence is confusing, pl

Thank you – this sentence was revised.

  1. Line 91-92: what do the authors mean with “amount of crops”? Please clarify.

Thank you - it was clarified.

  1. Line 115: “In the spring 2017 was transported…”. What was transported? Please specify.

Thank you – it was corrected.

  1. The use of commas and dots for decimals should be used consistently throughout the manuscript. For example, line 138 contains both “0.64” and “0,64”, and also Table 2 contains a comma. Please check and correct throughout the manuscript.

Thank you – it was checked and corrected.

  1. Table 2: the column headings of this table are not clear. Please use the previously introduced abbreviations (TB-0, TB-1, LH-0 and LH-1) as column headings.

Thank you – it was corrected

  1. Table 2: are the contents of Ntot, P, K, Ca, Mg, Na, and TC expressed per fresh weight or per dry weight? Please specify.

Thank you – it was corrected.

  1. Please check the consistent style of the tables. In tables 2-5, part of the first row is underlined and sometimes in bold.

Thank you – it was corrected.

  1. In the manuscript, sometimes total carbon is expressed by TC (e.g., in section 2.4) and sometimes by Ctot (e.g., in section 3.3). Is there a difference between the two?

It was corrected.

  1. The authors show that the application of MMDP generally improves the soil properties such as SBC, CEC, and BS. However, in the second year study this did not translate in a higher yield for lupine. Why not? Please clarify.

It was clarified.

I do trust that I follow all your remarks. Once more thank you!!

Reviewer 2 Report

Dear Authors,

You have done a good study and the manuscript needs a thorough  grammar check to further improve the readability. Some minor revisions are suggested from my side.

Regards

Author Response

Dear Reviewer,

Thank you very much for your review and your appreciation of our efforts. Grammar and spelling correction have been made.

With regards

Reviewer 3 Report

The main idea of this study is clear but there are some questions and discussion need to be clarified as the following.

1. The lines 11 to 16 of the ” Abstract ” are too long to be included in the abstract. The line 14 “C`H 4” was Clerical error.

2. In line 17, the name of “microorganism strains” should be written.

3. What are LH and TB in line 18? What does SPAD refer to in line 21? The full name and abbreviation should be stated for the first occurrence.

4. The expression of "minerals - mainly nitrogen" in line 47 is inaccurate.

5. The testing methods for “Characteristics of the soil” in table 1 should be indicated.

6. The description in lines 427 to 444 is more suitable for introduction than for discussion, which should be closely combined with the result analysis and references.

Author Response

Dear Reviewer,

Thank you very much for your competent review.

Please find our response below:

The lines 11 to 16 of the ” Abstract ” are too long to be included in the abstract. The line 14 “C`H 4” was Clerical error.

Thank you – corrections were made.

In line 17, the name of “microorganism strains” should be written.

It was done

  1. What are LH and TB in line 18? What does SPAD refer to in line 21? The full name and abbreviation should be stated for the first occurrence.

SPAD literally means unit of  Soil Plant Analysis Development and it created by company Konica Minolta and it is generally accepted as a unit of plant measurement of greenness. We would like to suggest leaving this abbreviation which is generally accepted in measurements in plant physiology.

  1. The expression of "minerals - mainly nitrogen" in line 47 is inaccurate.

Thank you – it was corrected.

  1. The testing methods for “Characteristics of the soil” in table 1 should be indicated.

For the clarity of the table we are suggesting leaving it in the present form because all methods have been described in the subsection 2.4.1 Soil analyses lines 160-188.

  1. The description in lines 427 to 444 is more suitable for introduction than for discussion, which should be closely combined with the result analysis and references.

Thank you for this remark -  in one aspect our intention was to show the problem in wider perspective but on the other hand you are absolutely right and sentences with descriptions were edited.

Once more thank you very much for your review.

With regards